# Metabolic Profiling of Pregnant Women with Obesity: An Exploratory Study in Women at Greater Risk of Gestational Diabetes

**DOI:** 10.3390/metabo12100922

**Published:** 2022-09-29

**Authors:** Ola F. Quotah, Lucilla Poston, Angela C. Flynn, Sara L. White

**Affiliations:** 1Department of Women and Children’s Health, School of Life Course and Population Sciences, King’s College London, 10th Floor North Wing, St Thomas’ Hospital, Westminster Bridge Road, London SE1 7EH, UK; 2Department of Clinical Nutrition, Faculty of Applied Medical Science, King Abdulaziz University, Jeddah 999088, Saudi Arabia; 3Department of Nutritional Sciences, School of Life Course and Population Sciences, King’s College London, Franklin-Wilkins Building, 150 Stamford Street, London SE1 9NH, UK

**Keywords:** gestational diabetes, maternal obesity, targeted metabolome, biomarkers

## Abstract

Gestational diabetes mellitus (GDM) is one of the most prevalent obstetric conditions, particularly among women with obesity. Pathways to hyperglycaemia remain obscure and a better understanding of the pathophysiology would facilitate early detection and targeted intervention. Among obese women from the UK Pregnancies Better Eating and Activity Trial (UPBEAT), we aimed to compare metabolic profiles early and mid-pregnancy in women identified as high-risk of developing GDM, stratified by GDM diagnosis. Using a GDM prediction model combining maternal age, mid-arm circumference, systolic blood pressure, glucose, triglycerides and HbA1c, 231 women were identified as being at higher-risk, of whom 119 women developed GDM. Analyte data (nuclear magnetic resonance and conventional) were compared between higher-risk women who developed GDM and those who did not at timepoint 1 (15^+0^–18^+6^ weeks) and at timepoint 2 (23^+2^–30^+0^ weeks). The adjusted regression analyses revealed some differences in the early second trimester between those who developed GDM and those who did not, including lower adiponectin and glutamine concentrations, and higher C-peptide concentrations (FDR-adjusted *p* < 0.005, < 0.05, < 0.05 respectively). More differences were evident at the time of GDM diagnosis (timepoint 2) including greater impairment in β-cell function (as assessed by HOMA2-%B), an increase in the glycolysis-intermediate pyruvate (FDR-adjusted *p* < 0.001, < 0.05 respectively) and differing lipid profiles. The liver function marker γ-glutamyl transferase was higher at both timepoints (FDR-adjusted *p* < 0.05). This exploratory study underlines the difficulty in early prediction of GDM development in high-risk women but adds to the evidence that among pregnant women with obesity, insulin secretory dysfunction may be an important discriminator for those who develop GDM.

## 1. Introduction

Pregnancy is associated with major physical and metabolic alterations, required to meet increased maternal metabolic demands and permit fetal development [1]. Obesity in pregnancy impacts metabolic adaptations and heightens risk of adverse pregnancy outcomes, most notably gestational diabetes mellitus (GDM) [2]. The physiological increase in insulin resistance observed in late pregnancy is further exacerbated in women with obesity causing an increase in postprandial circulating glucose, amino acids and lipids [3].

The pathogenesis of GDM in pregnant women with obesity is complex, with contribution of exaggerated insulin resistance, dyslipidaemia, reduced insulin secretion, increased inflammatory cytokines, and changes in amino acids, fatty acids, ketone bodies and adipokines [4,5]. As GDM has been shown to be associated with pregnancy complications [6,7] and metabolic diseases later in life [8,9], a better understanding of the pathophysiology of GDM would enable early detection and targeted intervention, particularly in higher-risk women who would benefit the most.

Metabolomic analysis is an approach which combines analytical chemistry with computational techniques to investigate large numbers of metabolites [10]. Previous studies have utilised metabolomics to identify biomarkers for early risk stratification of women for later development of GDM [11,12,13,14]. Several differences in the metabolic profiles of women who are obese with GDM and those without GDM at the time of and prior to diagnosis have been reported [4,15]. A previous study undertaken by White et al. [4] described metabolic profiles of 1303 pregnant women with obesity who had taken part in the UPBEAT study. Numerous analytes were measured including 147 from a targeted nuclear magnetic resonance (NMR) metabolome. Increased lipids and lipoprotein constituents in very low-density lipoprotein (VLDL) subclasses and higher triglycerides enrichment across lipoprotein particles were detected in women with obesity who ultimately developed GDM. Women with GDM had elevated levels of branched-chain amino acids (BCAA) and aromatic amino acids (ACAA). Differences in glucose homeostasis, fatty acids, ketone bodies, adipokine, liver and inflammatory marker profiles were also observed in GDM women compared with those without GDM. These differences were observed at least 10 weeks prior and at the time of GDM diagnosis [4]. Similar findings were reported by Mokkala et al. [15] who investigated early metabolite changes in 357 pregnant women with obesity prior to GDM diagnosis to determine metabolites associated with the development of GDM. The study identified 78 divergent lipid metabolites in early pregnancy between women who developed GDM (n = 82) and those who did not (n = 275).

Several tools have been developed to identify pregnant women at higher-risk of developing GDM [16,17,18,19,20,21,22]. Weight or BMI is invariably included in these prediction models, limiting their predictive capacity in pregnant women with obesity. However, one tool, developed by White et al. combines biochemical and clinical factors to facilitate early identification of pregnant women with obesity at increased risk of developing GDM [23]. This pragmatic prediction tool, constructed using logistic regression, consisting of maternal age, blood pressure, mid-arm circumference, glucose, haemoglobin A1c (HbA1c) and triglycerides [23] was selected to be translated for use in a clinical setting [24]. Participant predicted risks were calculated using the model algorithm and converted into binary categories using a ≥35% risk threshold; the tool identifies a group of high-risk women with obesity of whom approximately 50% (positive predictive value) later develop GDM with a negative predictive value of approximately 80% [23].

The aim of this study was to explore which metabolic differences (analytes measured by NMR and conventional platforms) exist between individuals identified as higher-risk via the prediction tool but stratified by the development of GDM. We hypothesised that the identification of such differences might lead to a better understanding of the pathophysiological pathways to hyperglycaemia which could ultimately facilitate the development of more accurate methods of GDM prediction.

## 2. Materials and Methods

### 2.1. Study Design and Population

This was a secondary analysis of data collected from the UK Pregnancies Better Eating and Activity Trial (UPBEAT; isrctn.org registration number). The UPBEAT multicentre RCT was a behavioural intervention in pregnant women with obesity aimed to prevent GDM and delivery of large-for gestational-age (LGA) infants [24]. From 2009 to 2014, 1555 women were recruited and randomised to receive either behavioural intervention or standard antenatal care. Women were eligible to participate in the trial if they were at least 16 years old or above, had a BMI ≥ 30 kg/m^2^ and a singleton pregnancy between 15^+0^ and 18^+6^ gestation. Women with pre-existing medical conditions, unable to give informed consent and those who had been prescribed metformin were excluded. Ethical approval was obtained by the NHS Research Ethics Committee (UK Integrated Research Application System, reference 09/H0802/5) [24].

As the UPBEAT trial did not show a significant difference in the primary outcomes of GDM and LGA infants between treatment groups, the trial participants were treated as a cohort for the purposes of this study. However, to adjust for any intervention effect on the analytes, intervention allocation was included in the model at timepoint 2 (between 23^+2^ to 30^+0^ weeks). This was a complete case analysis including high-risk women identified by the prediction tool (n = 231), who had a diagnostic oral glucose tolerance test (OGTT), provided blood samples at the UPBEAT baseline (15^+0^ to 18^+6^ weeks gestation, timepoint 1) and OGTT (23^+2^ to 30^+0^ weeks, timepoint 2) visits with complete analyte data at both timepoints. Thirty-four women from one centre were excluded as their blood samples were not obtained at the time of the first fasting blood sample of OGTT. Of note, the prediction tool was developed at timepoint 1 in this cohort [23].

### 2.2. Study Procedures

For the main trial, only OGTTs that were undertaken between 27^+0^ and 28^+6^ were included. However, in this study a clinically pragmatic approach was adopted to include all OGTTs at 23^+2^ to 30^+0^ weeks. The OGTT consisted of a fasting glucose measurement (≥10 h), followed by glucose measurements 1-h and 2-h after drinking the glucose load (75 grams). During the OGTT, a research sample was collected at the time of the first fasting blood sample. A random blood test (non-fasted) was also obtained at study entry (timepoint 1). Blood samples were kept on dry ice and processed within 2 h with subsequent storage at −80 °C. Metabolite analyses were undertaken by technicians blinded to the participant data.

### 2.3. Diagnosis of GDM

GDM was diagnosed based on the International Association of Diabetes and Pregnancy Study Groups (IADPSG) criteria as one or more of a fasting glucose ≥ 5.1 mmol/L, 1 h ≥ 10.0 mmol/L and 2 h ≥ 8.5 mmol/L, following a 75 -g oral glucose load [25].

### 2.4. Metabolic Profiling

A total of 163 analytes were evaluated at both timepoints combining an NMR metabolome and conventional laboratory assays. Samples (serum) were sent to Nightingale Health, Finland (research.nightingalehealth.com/, accessed 10 August 2022) for metabolomic analysis, while conventional biochemical analyses were performed by members of the UPBEAT study team at Glasgow University. Analyses were performed in two batches (March 2015 and August 2015).

The NMR platform has been extensively applied in previous epidemiological studies [26,27,28,29,30] and there are no discernable batch effects [31,32]. The methodology is briefly described in Appendix A. The platform allows a comprehensive quantification of over 100 primary metabolites in addition to >100 derived measures, in either serum or EDTA plasma. This includes measuring lipoprotein classes, constituent lipids, fatty acids, sphingolipids, inflammatory markers and numerous low-molecular-weight metabolites (LMWM) such as amino acids, glycolysis-related metabolites and ketone bodies.

For analysis of lipids, lipoprotein particles were measured, including VLDL subdivided into six subclasses (extremely large, very large, large, medium, small, very small), intermediate-density lipoprotein (IDL), low-density lipoprotein (LDL) subdivided into three subclasses (large, medium, small) and high-density lipoprotein (HDL) subdivided into four subclasses (very large, large, medium, small). The constituents within each lipoprotein particle type (triglycerides, total cholesterol, non-esterified cholesterol and cholesteryl ester levels and phospholipid concentrations) were quantified. Fatty acids, amino acids, glycolysis-related metabolites, ketone bodies and inflammatory markers were also analysed.

Conventionally measured biomarkers (n = 16) were selected based on associations with type 2 diabetes (T2DM), GDM or insulin resistance (Appendix A) [33]. The markers examined included four glucose-homeostasis markers (HbA1c, fructosamine, insulin, C-peptide), markers of inflammation and endothelial dysfunction (high-sensitivity C-reactive protein (hs-CRP), IL−6, tissue plasminogen activator (tPA) antigen and ferritin), markers of liver function (alanine aminotransferase (ALT), aspartate aminotransferase (AST), γ-glutamyl transferase (gGT), and sex hormone binding globulin (SHBG)), adipokines (adiponectin, leptin), vitamin D and human placental lactogen (hPL).

Vitamin D, hPL and HbA1c were evaluated at timepoint 1 only, while insulin resistance indices (HOMA2-IR, updated HOMA of insulin resistance; HOMA2-%S, updated HOMA of insulin sensitivity; HOMA2-%B, updated HOMA of steady-state β-cell function) were evaluated only at timepoint 2. Glucose measurements within the OGTT were not considered in the analysis at timepoint 2 as these are fundamental to GDM diagnosis. Other than that, all analytes were measured at both timepoints.

### 2.5. Data Collection

For each UPBEAT participant, sociodemographic, clinical and anthropometric data were collected at timepoint 1. Data included age (years), BMI (kg/m^2^), parity (nulliparous, multiparous), ethnicity (African, African-Caribbean, South Asian, European, other), education (degree, A level or equivalent, vocational qualification, GCSE or equivalent, none), smoking status (non-smoker, current smoker, ex-smoker) and mid-arm circumference (cm). Blood pressure (mmHg) was also recorded. Medical history included family history of diabetes (type 1, type 2) and previous history of GDM (yes, no).

Maternal and neonatal outcome information was extracted from the UPBEAT database and included GDM (defined by IADPSG); pre-eclampsia (defined as systolic blood pressure ≥ 140 mm Hg, diastolic blood pressure ≥ 90 mm Hg, or both, on at least two occasions 4 h apart, plus proteinuria (≥300 mg/24 h) or spot urine protein:creatinine ratio (≥30 mg/mmol creatinine), or urine dipstick protein (≥2+)); caesarean section (elective, emergency, pre-labour, in labour); gestational age at delivery (weeks); neonatal sex (male/female); birthweight (g) and LGA infant (≥90th birthweight customised centile). As part of the metabolomic analysis, analytes were checked for variation for gestational age at sampling and converted into corrected centiles as necessary.

### 2.6. Statistical Analysis

Analytes were checked for normality visually using normality plots; data that were not normally distributed were log transformed. Normally distributed continuous variables were presented as mean and standard deviation (SD) or median and interquartile range (IQR), if the distribution was skewed. For categorical and binary variables, the number (N) and percentage (%) were presented. Summary statistics were compared between high-risk women who developed GDM and those who did not develop GDM using Student’s *t* test or Mann-Whitney tests for continuous data or Chi-Square tests for categorical and binary data as appropriate.

Analyte data were compared between high-risk women who developed GDM and those who did not, at timepoint 1 and at timepoint 2 using multivariate regression analyses adjusted for potential confounding factors including maternal BMI, parity, ethnicity, age, education and neonatal sex. In order to adjust for any intervention effect on the analytes, treatment group was additionally included in the model at timepoint 2. SD difference was reported to allow comparison between multiple measures that were recorded in different units.

A false discovery rate (FDR) approach [34] was used to decrease the probability of false-positive results and to minimise the effects of multiple comparisons. Statistical significance was identified if the FDR-corrected *p* value was <0.05. Statistical analyses were performed using Stata software, version 16.0 (StataCorp, College Station, TX, USA).

## 3. Results

### 3.1. Participant Characteristics

Of the 1555 women who took part in UPBEAT [24], 231 were identified as being at a higher-risk of developing GDM using the prediction tool, of whom 119 women (51.5%) actually developed GDM. Participant characteristics are shown in Table 1. In high-risk women who developed GDM, there was a significantly higher rate of LGA infants and infants were delivered at an earlier gestational age. Non-fasting glucose concentrations and HbA1c were higher at timepoint 1 in those who developed GDM (both *p* < 0.05).

### 3.2. Metabolic Profiles

Differences in metabolic profiles between high-risk women who developed GDM and those who did not were explored after adjustment for potential confounders at the two timepoints. Associations between analytes and GDM (difference in SD) at both timepoints are shown in Figure 1, Figure 2 and Figure 3, and SD differences are noted in Appendix A. Concentrations of the analytes at timepoint 1 and 2 are shown in Appendix A.

#### 3.2.1. Timepoint 1 (15^+0^ to 18^+6^ Weeks Gestation)

Few metabolic differences were identified at timepoint 1 in this high-risk subgroup of women (Figure 1a, Figure 2a and Figure 3a). Phospholipid and free cholesterol in small HDL were positively associated with the development of GDM, while phospholipid concentrations in very large HDL particles were negatively associated with GDM. There was a negative association between developing GDM and the ratio of linoleic acid to total fatty acids. The development of GDM was positively associated with glucose and C-peptide levels and negatively associated with glutamine concentrations with no further differences evident in amino acids and ketone bodies prior to GDM diagnosis between the two groups. Adiponectin, SHBG and IL−6 concentrations were inversely associated with GDM whereas gGT was positively associated. Metabolite absolute values and SD difference at timepoint 1 are shown in Appendix A.

#### 3.2.2. Timepoint 2 (OGTT, 23^+2^ to 30^+0^ Weeks Gestation)

Numerous differences between high-risk women who developed GDM and those who did not develop GDM were found at the time of the diagnostic OGTT (Figure 1b, Figure 2b and Figure 3b). GDM was negatively associated with total lipids in IDL and LDL subclasses; total cholesterol; total cholesterol in very small VLDL, IDL, all LDL subclasses, very large HDL, small HDL and HDL3; total esterified cholesterol; cholesterol esters in small VLDL, very small VLDL, very large/small HDL, IDL and LDL subclasses; total free cholesterol and free cholesterol in very small VLDL, IDL, LDL subclasses and large/very large HDL. Total phospholipids; phospholipids in very large HDL, very small VLDL, IDL, LDL and all its subclasses; triglycerides in medium LDL; total cholines; sphingomyelins as well as phosphatidylcholine and other cholines followed a similar pattern. Phospholipid and free cholesterol in small HDL and triglycerides in both chylomicrons and extremely large VLDL were positively associated with GDM. Among the fatty acids, linoleic acid, omega−6, polyunsaturated fatty acids (PUFA) and the proportion of omega−6, linoleic acid and PUFA to total fatty acids were negatively associated with GDM, whereas the ratio of monounsaturated fatty acids (MUFA) and saturated fatty acids (SFA) to total fatty acids both had a positive association with the development of GDM. Fructosamine, glycolysis intermediate pyruvate, BCAAs (valine, leucine and isoleucine) and ACAAs (tyrosine) along with acetoacetate and gGT concentrations were all positively correlated with GDM, while insulin indices assessed by HOMA scores, revealed a negative association between β-cell function with GDM diagnosis, but no difference in insulin resistance indices. Furthermore, GDM was negatively associated with adiponectin concentrations. Metabolite absolute values and SD difference at timepoint 2 are shown in Appendix A.

## 4. Discussion

To our knowledge, this is the first attempt to examine maternal metabolomic profiles in high-risk women with obesity who were identified using a novel multivariable prediction tool for GDM. Of note, higher-risk women were stratified by the development of GDM and those who developed GDM showed lower adiponectin and glutamine concentrations, and higher C-peptide concentrations at timepoint 1, greater impairment in β-cell function (indicated by HOMA2-%B), increased glycolysis-intermediate pyruvate and differing lipid profiles at timepoint 2, with elevated gGT levels at both timepoints. The differences identified in this study may facilitate improved understanding of the underlying pathophysiological processes leading to hyperglycaemia and therefore better early recognition or prediction of those who will most likely develop GDM.

### 4.1. Timepoint 1 (Non-Fasting Sample at Baseline; 15^+0^ to 18^+6^ Weeks’ Gestation)

Comparison between variables demonstrated statistically significant but marginal clinical differences in glucose and HbA1c between high-risk women who developed GDM and those who did not. Fasting plasma glucose (FPG) (≥5.1 mmol/L) and HbA1c (≥5.7%) as single risk factors during early pregnancy have been shown to be useful in early risk stratification of GDM [35,36]. However, as shown in the present study, when using glucose and HbA1c as single predictors (unadjusted) it was not possible to discriminate between high-risk women who later developed GDM and those who did not. Of note, both these biomarkers were included in the prediction model and due to the nature of the multivariable risk-stratified approach, some variables would have been more influential than others for a particular individual [23]. C-peptide concentrations, which may predict impaired insulin action particularly if associated with higher plasma glucose [37] were higher in the GDM women.

In this high-risk cohort, glutamine concentrations were lower at timepoint 1 in women who later developed GDM compared to those that did not. There is a lack of detailed cellular research assessing the direct effects of glutamine deprivation on pancreatic β-cells metabolism and function. However, it was recently reported in an in vitro study of isolated human islets that decreased glutamine concentration is associated with severe β-cell dysfunction as evidenced by reduced insulin synthesis and secretion. The finding was accompanied by signs of both cellular oxidative and endoplasmic reticulum stress as well as enhanced lipotoxicity [38]. In non-pregnant populations, reduced glutamine concentrations have been observed in obese individuals with prediabetes [39]. Taken together, these studies imply that the low glutamine concentrations observed at timepoint 1 in the women who developed GDM might reflect both lower insulin secretion as well as increased insulin resistance and may be indicative of an underlying mechanism leading to GDM in this high-risk group.

In the present study, those high-risk women who developed GDM had lower concentrations of SHBG at timepoint 1 only [40]. SHBG is inversely associated with insulin resistance [41] and development of GDM [42,43], however, the mechanism for this association remains unclear. The plasma adiponectin concentration was also significantly lower in high-risk women who developed GDM compared to women who did not. Adiponectin is believed to play a major role in regulating glucose and lipid metabolism and attenuating insulin resistance [44] and has been identified as a valuable predictor of GDM, regardless of baseline BMI, in this, and other cohorts [4,45,46]. Evidence from in vivo and in vitro experimental studies have shown that the predominant impact of adiponectin on the pancreas includes the stimulation of β-cell function, survival and proliferation [47]. It is suggested that adiponectin increases insulin cell content and enhances glucose-stimulated insulin secretion by inducing activation of peroxisome proliferator-activated γ receptors [48]. In addition, as recently reported, high molecular weight adiponectin is positively associated with insulin secretion (evaluated using HOMA-β%), with or without the adjustment for insulin resistance [49]. These findings suggest that hypoadiponectinemia might not only reflect insulin resistance, but also β-cell dysfunction, both contributing to the pathophysiology of GDM.

Lipid and fatty acid profiles amongst identified high-risk women with obesity stratified by GDM development were similar early in pregnancy. This is in contrast to a previous comparison early in pregnancy between all-risk women with obesity who did and did not develop GDM, where many lipid differences were evident at timepoint 1 [4,15,50]. It also contrasts to timepoint 2, when many differences were found (see below). This finding likely reflects fewer differences across an identified high-risk group prior to the development of hyperglycaemia that then drives the dyslipidaemia.

### 4.2. Timepoint 2 (Fasting Sample at Time of Diagnostic OGTT; 23^+2^ to 30^+0^ Weeks’ Gestation)

At the time of GDM diagnosis, the higher-risk group of women who developed GDM demonstrated a dysregulated lipid profile, increased branched-chain and aromatic amino acids and differing fatty acid, ketone body, adipokine and liver marker profiles compared with women without GDM. Some of these differences were in line with previous reports [4,15,50], and reflect closely what had been found previously in the whole UPBEAT cohort between GDM and non-GDM women [4].

Of note, β-cell dysfunction appears to be an important differentiating factor between development of GDM and non-GDM in women identified as being at risk in early pregnancy; high-risk women with GDM showed greater impairment of β-cell function at diagnosis (assessed by HOMA2-%B indices), reflecting lower insulin secretory function. In most previous studies, high BMI has been associated with abnormal insulin resistance, yet these data suggest that insulin secretory dysfunction also plays a crucial role in the pathogenesis of GDM in obese women. This supports findings from a recent study of 40 GDM women from Bangladesh with increased BMI (≥23 kg/m^2^) where GDM was associated with a combination of both insulin resistance and inadequate insulin secretion [51]. Further evidence, from postpartum studies, also suggests that β-cell function is the most significant factor for the progression to future diabetes [52,53,54], particularly in the presence of obesity/insulin resistance and a history of previous GDM [55]. Furthermore, indicative of different ‘at risk’ subtypes, our group, using metabolic profiling, proposed different pathophysiological processes in 71 GDM women with obesity when stratified by different treatment modalities used to achieve glycaemic control; GDM women who were ultimately treated with insulin displayed a more insulin resistant profile compared to those who were managed with diet suggesting insulin insufficiency in this latter group [56].

Further possible evidence of insulin secretory dysfunction was the increased concentration of a glycolysis-intermediate, pyruvate, in high-risk women with GDM compared to those without. Mitochondrial metabolism of pyruvate is critical for insulin secretion and there is evidence of implicated mitochondrial pyruvate uptake in association with T2DM [57] although, the association between pyruvate levels and β-cell function is not fully understood [58]. Decreased mitochondrial metabolism of pyruvate by pyruvate dehydrogenase and pyruvate carboxylase would reduce signaling molecules thought to be critical for regulating insulin secretion [58]. Therefore, the higher pyruvate levels could result from increased glucose metabolism per se, but might also represent decreased transport into, or metabolism in, the mitochondria which might impact on insulin release and GDM development.

Serum gGT levels were elevated in higher-risk GDM pregnancies at both timepoints. Elevated gGT is considered to be a marker of increased oxidative stress [59] and has been associated with subsequent risk of GDM [60]. The mechanism of this association remains unclear. One mechanistic explanation is that β-cells have a low antioxidant capacity, which make these cells more prone to oxidative stress, and an imbalance in redox homeostasis between reactive oxygen species and antioxidant capacity induces β-cell de-differentiation, reducing insulin secretion by pancreatic β-cells [61]. Cellular oxidative stress is also proposed to affect insulin sensitivity by inducing insulin signal disruption and adipocytokine dysregulation, thus contributing to diabetes development [62,63,64].

Early in pregnancy, there is a progressive increase in maternal insulin secretion to compensate for the physiological changes in insulin sensitivity [65], which is proposed to increase the risk of developing GDM in women with pre-existing insulin resistance, especially in those with obesity. While the prediction model utilised in this study to identify women at risk of GDM used variables that are insulin-resistance focused, it is interesting to note that further differentiation between these high-risk women into those who did and did not develop GDM highlights a number of insulin secretory markers when measured at the time of GDM diagnosis. These data imply that the inclusion of biomarkers of insulin secretory dysfunction measured early in pregnancy into a prediction tool could lead to a more discriminatory test that differentiates between women who subsequently do and do not develop GDM.

### 4.3. Strengths and Limitations

This study has several strengths. Only high-risk women who had been identified using a novel prediction tool were included. We believe there has been no previous attempt to further refine risk assessment of GDM by exploring differences in metabolic profiles between high-risk women who did and did not develop GDM using this approach. Furthermore, the rich UPBEAT dataset, providing substantial demographic data for all women enabled adjustment for relevant potential confounding factors in the statistical analysis. One limitation in this analysis is the relatively small sample size, and validation in another larger study is recommended.

## 5. Conclusions

This study has improved the understanding of different metabolic pathways in GDM amongst an identified higher-risk cohort of pregnant women as assessed by NMR spectroscopy and conventional biochemical assays. We have reported novel metabolic profile differences specific to GDM development in higher-risk women. Although obesity and measures of insulin resistance were similar between GDM and non GDM groups, notable highlighted metabolic abnormalities were seen in insulin secretion in the GDM group. These data indicate that impairment in insulin secretory capacity of pancreatic β-cells might be a key discriminator of GDM development in women at higher risk. The aetiology of GDM remains elusive, and no single biomarker, or predictive model has yet been approved for use or integrated into clinical practice. This study has highlighted prognostic factors which might facilitate risk assessment. Future studies should focus on evaluating other factors beyond conventional clinical and metabolic risk factors that do not fully explain this excess risk.

In conclusion, our data provide an overview of metabolite alteration from a higher-risk antenatal population identified using a multivariable model. Although exploratory in nature, this study raises the hypothesis that several metabolites, related to defective β-cell secretory function including adiponectin, glutamine, insulin secretory index HOMA-β%, pyruvate and gGT may contribute to improved GDM risk assessment in women with obesity. In turn, this would identify those women most likely to benefit from a targeted intervention.

## Figures and Tables

**Figure 1 metabolites-12-00922-f001:**
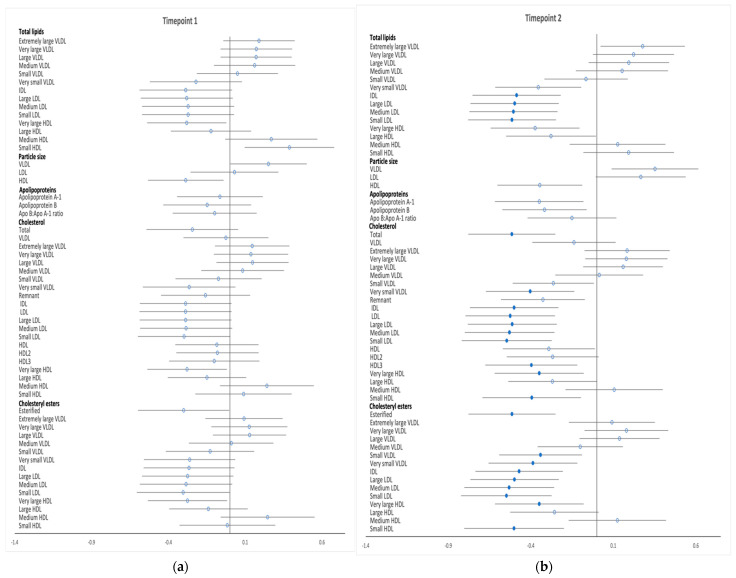
Differences in total lipids in all lipoprotein subclasses, particle size, apolipoproteins, free cholesterol and cholesteryl esters between high-risk women who developed GDM and those who did not develop GDM at timepoints 1 (**a**) and timepoint 2 (**b**). Data points show the SD difference at baseline (timepoint 1) and at the time of OGTT (timepoint 2). Positive associations with GDM are shown to the right, negative associations are shown to the left. Closed blue circles represent FDR-corrected *p* values of <0.05.

**Figure 2 metabolites-12-00922-f002:**
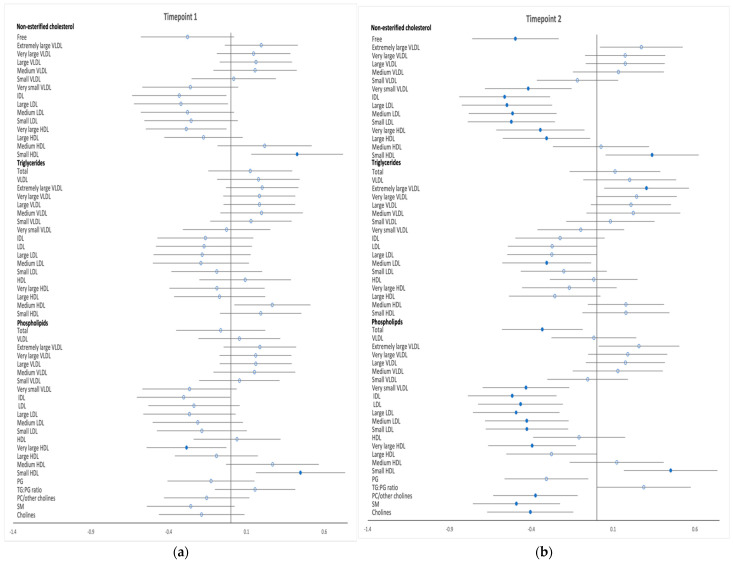
Differences in non-esterified cholesterol, triglycerides and phospholipids between high−-risk women who developed GDM and those who did not develop GDM at timepoints 1 (**a**) and timepoint 2 (**b**). Data points show the SD difference at baseline (timepoint 1) and at the time of OGTT (timepoint 2). Positive associations with GDM are shown to the right, negative associations are shown to the left. Closed blue circles represent FDR corrected *p* values of <0.05. PC, phosphatidylcholines; PG, phosphoglycerides; SM, sphingomyelins; TG:PG, triglycerides: phosphoglyceride.

**Figure 3 metabolites-12-00922-f003:**
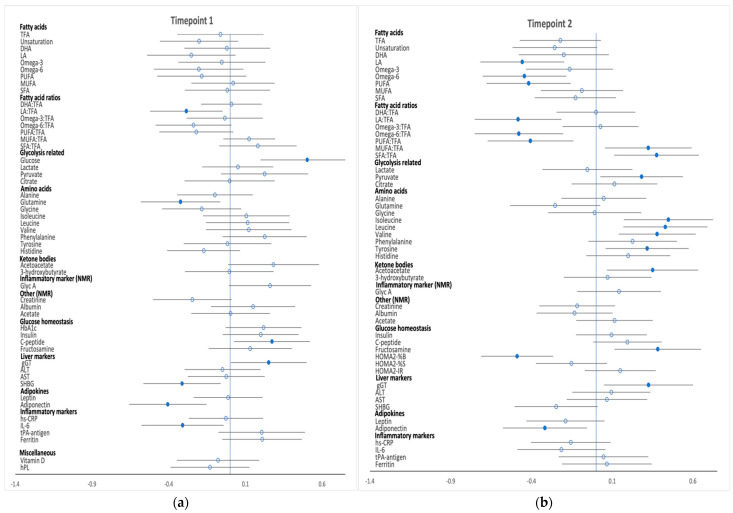
Differences in fatty acids, glycolysis-related metabolites, amino acids, ketone bodies and inflammatory and other markers between high-risk women who developed GDM and those who did not develop GDM at timepoints 1 (**a**) and timepoint 2 (**b**). Data points show the SD difference at baseline (timepoint 1) and at the time of OGTT (timepoint 2). Positive associations with GDM are shown to the right, negative associations are shown to the left. Closed blue circles represent FDR-corrected *p* values of <0.05. DHA, docosahexaenoic acid, 22:6; LA, linoleic acid, 18:2; MUFA monounsaturated fatty acids, 16:1, 18:1; PUFA polyunsaturated fatty acids; SFA Saturated fatty acids; TFA, total fatty acid.

**Table 1 metabolites-12-00922-t001:** Characteristics of high-risk women identified by the prediction tool who developed GDM and those who did not develop GDM.

Characteristics	GDM (n = 119)	No GDM (n = 112)	*p* Value ^a^
**Ethnicity**			0.41
African	23 (19.3)	13 (11.6)	
African-Caribbean	11 (9.2)	8 (7.1)	
South Asian	9 (7.6)	10 (8.9)	
European	64 (53.8)	72 (64.3)	
Other	12 (10.1)	9 (8.0)	
**Parity**			0.97
Nulliparous	46 (38.7)	43 (38.4)	
Multiparous	73 (61.3)	69 (61.6)	
**Education level**			0.08
Degree	50 (42.0)	47 (42.0)	
A level (or equivalent)	16 (13.4)	23 (20.5)	
Vocational qualification	36 (30.2)	18 (16.1)	
GCSE (or equivalent)	12 (10.1)	17 (15.2)	
None	5 (4.2)	7 (6.2)	
**Smoking status**			0.53
Non-smokers	81 (68.1)	83 (74.1)	
Smokers	7 (5.9)	4 (3.6)	
Ex-Smokers	31 (26.0)	25 (22.3)	
**Maternal weight (kg)**			
At timepoint 1	101.9 (20.1)	102.5 (15.1)	0.82
At timepoint 2 ^b^	105.5 (20.5)	105.9 (15.5)	0.86
Previous GDM ^b^	6 (5.0)	2 (1.8)	0.24
**Family history**			
T1DM	2 (1.7)	5 (4.5)	0.22
T2DM	42 (33.1)	36 (26.1)	0.41
**Pregnancy outcomes**			
Pre-eclampsia ^b^	6 (5.2)	10 (9.3)	0.22
Caesarean section (all) ^b^	54 (45.8)	49 (44.5)	0.85
Caesarean section (emergency) ^b^	22 (18.6)	20 (18.2)	0.93
Preterm delivery (<37 weeks) ^b^	1 (0.9)	1 (0.9)	0.95
LGA (90 th customised centile) ^b^	16 (13.6)	5 (4.5)	**0.02 ***
Birthweight (g) ^b^	3400 (3120–3720)	3466 (3070–3785)	0.44
**Gestational age (weeks)**			
At timepoint 1	16.0 (15.0–16.0)	16.0 (15.0–17.0)	0.33
At timepoint 2 ^b^	27.0 (27.0–28.0)	27.0 (27.0–28.0)	0.82
At delivery ^b^	39.0 (38.3–39.8)	40.1 (38.8–40.8)	**<0.001 ***
**Treatment group**			0.48
Control	64 (53.8)	55 (49.1)	
Intervention	55 (46.2)	57 (50.9)	
**GDM prediction model components (at timepoint 1)**			
Maternal age (years)	32.6 ± 4.3	33.6 ± 5.3	0.13
BMI (kg/m^2^)	36.7 (33.8–40.4)	36.9 (34.0–41.3)	0.78
Systolic blood pressure (mmHg)	123.4 ± 11.1	122.3 ± 11.2	0.45
Mid-arm circumference (cm)	39.0 (36.0–41.0)	38.0 (35.5–41.5)	0.46
HbA1c (mmol/mol) ^c^	32.9 ± 3.9	31.5 ± 3.4	**0.006 ***
Glucose (mmol/l) ^c^	5.5 ± 1.0	5.1 ± 0.7	**0.001 ***
Triglycerides (mmol/l) ^c^	1.0 ± 0.4	1.0 ± 0.4	0.89

Data are shown as number (%), mean ± SD or median (interquartile range). ^a^ *p* value from χ2, Student’s *t* test or Mann-Whitney test as appropriate. ^b^ Missing data: history of GDM (n = 2), LGA (n = 3), pre-eclampsia (n = 8), caesarean section all/emergency (n = 3), preterm birth (n = 9), birthweight (n = 3), gestational age at point 2 (n = 1), maternal weight (n = 2), gestational age at delivery (n = 3). ^c^ Analytes measured using a non-fasting sample. * *p* < 0.05. BMI, body mass index; T1DM, type 1 diabetes mellitus; T2DM, type 2 diabetes mellitus; LGA, large-for-gestational-age.

## Data Availability

The data presented in this study are available in Appendix A.

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
