# Peer review of "Metabolic Profiling of Pregnant Women with Obesity: An Exploratory Study in Women at Greater Risk of Gestational Diabetes"

_metabolites, 2022, doi:10.3390/metabo12100922_

Round 1

Reviewer 1 Report

This study attempts to differentiate metabolomic profiles between GDM versus non-GDM cases among a total cohort of women with obesity deemed to be at high risk of GDM based on other clinical factors. This is an important consideration given that obesity, defined by BMI, includes a wide array of metabolic phenotypes. However, this reviewer’s main concern relates to the use of the prediction model for determining risk of GDM in this cohort. The low sensitivity and moderate-high specificity of the prediction model developed by White et al. may not be the most reliable foundation on which to base the present analysis. At the very least, this should be highlighted as a limitation of the study.

Abstract

1.     Line 18-19: the way the aim is stated is somewhat confusing. Only by reading the manuscript the reader understands that you are using early pregnancy measures to predict GDM that develops later in pregnancy. The text currently implies that you are identifying women at high risk of developing early-pregnancy GDM, which is not true or plausible.

2.     Line 22-23: need to define what the comparison was based on – presumably those who developed GDM versus those who did not?

Introduction

3.     Line 70-81: after presenting the previous literature published on this topic, the authors need to outline more clearly how this study is advancing the science. White et al. developed the prediction model using accessible biomarkers in the clinical setting from this same prenatal cohort, but they also looked at NMR metabolites.

4.     Line 82-83: the authors state that “Metabolic biomarkers in pregnant women with obesity identified at higher-risk of developing GDM have not previously been examined”. However, this does not appear to be true based on the prior work of White et al (ref #23) who utilized the same cohort and tested the contribution of NMR metabolites to their prediction model for GDM risk. Although the approach is different between the two papers, the fact that White et al did not find added benefit from including metabolites in their model appears to undermine the objective for the current paper. The authors need to provide stronger justification for the rationale behind the study.

Materials and Methods

5.     Line 325-326: please briefly describe how the prediction tool operates to select those at high-risk of GDM and provide reference to the original study here.

6.     Line 333-336: was there any consideration for the effect of gestational age in the models, given that diagnosis of GDM was based on OGTTs performed over a wide time frame (23-30 weeks)?

7.     Line 351: please describe approaches for measuring the conventional biomarkers and also provide inter- and intra-assay CV values. This is particularly important given that assays and NMR were run in 2 separate batches.

Results

8.     Figures: Font size in figures 1 and 2 should be increased to improve readability. Please specify in the captions if the open blue circles represent uncorrected p-values?

Discussion

9.     Line 305-307: it should be noted that the prediction model for GDM risk on which this analysis is based has poor sensitivity and thus, many women from the UPBEAT trial who were indeed at risk for GDM were not included. Also, the specificity of the prediction tool is decent but could be higher, suggesting that some women could have been inappropriately classified as high risk for GDM.

Author Response

Reviewer 1 
We wish to thank the reviewer for their comments. We have addressed the comments below. 

Comments: 
Point 1: Line 18-19: the way the aim is stated is somewhat confusing. Only by reading the manuscript the reader understands that you are using early pregnancy measures to predict GDM that develops later in pregnancy. The text currently implies that you are identifying women at high risk of developing early-pregnancy GDM, which is not true or plausible.

Response 1: We have amended the aim in the abstract as suggested to make it clear that we compared metabolic profiles early in pregnancy to predict GDM that develops later in pregnancy (lines 21-22).

Point 2: Line 22-23: need to define what the comparison was based on – presumably those who developed GDM versus those who did not?

Response 2: We have added a definition of comparison between the two groups (lines 25-26).

Point 3: Line 70-81: after presenting the previous literature published on this topic, the authors need to outline more clearly how this study is advancing the science. White et al. developed the prediction model using accessible biomarkers in the clinical setting from this same prenatal cohort, but they also looked at NMR metabolites.

Response 3:  We have amended the introduction to increase the clarity of the study aim. As we hope is now clear, this study takes the earlier work by White et al. further by exploring whether analyte differences between women who were identified as high risk but did not develop diabetes differed in a meaningful way from women who were identified as high risk but did develop diabetes.

Point 4: Line 82-83: the authors state that “Metabolic biomarkers in pregnant women with obesity identified at higher-risk of developing GDM have not previously been examined”. However, this does not appear to be true based on the prior work of White et al (ref #23) who utilized the same cohort and tested the contribution of NMR metabolites to their prediction model for GDM risk. Although the approach is different between the two papers, the fact that White et al did not find added benefit from including metabolites in their model appears to undermine the objective for the current paper. The authors need to provide stronger justification for the rationale behind the study.

Response 4: We agree that the inclusion of this sentence was confusing and we have subsequently removed it. It is this specific subgroup of high-risk obese women that has not previously been examined and we hope that the text is now clearer. 

Point 5: Line 325-326: please briefly describe how the prediction tool operates to select those at high-risk of GDM and provide reference to the original study here.

Response 5: Thank you for noting this omission, we have enriched the description of the tool, slightly earlier than suggested but we hope in an appropriate place (lines 89-94). 

Point 6: Line 333-336: was there any consideration for the effect of gestational age in the models, given that diagnosis of GDM was based on OGTTs performed over a wide time frame (23-30 weeks)? 

Response 6:  The diagnosis of GDM was accepted whenever the diagnostic OGTT was carried out in the end 2nd/early 3rd trimester regardless of gestational age. However, the effect of gestational age on the biomarkers was considered – they were checked for variation and transformed into centiles if required (lines 222-224).

Point 7: Line 351: please describe approaches for measuring the conventional biomarkers and also provide inter- and intra-assay CV values. This is particularly important given that assays and NMR were run in 2 separate batches.

Response 7: Candidate biomarker assessment was carried out in Glasgow University with strict quality control with the manufacturer’s calibrators and quality controls. CVs were monitored for reproducibility. The supplementary information contains details of the platforms used and the CVs for each analyte. The metabolomic analysis also undergoes rigorous quality control described in more detail in the following reference (Quantitative serum nuclear magnetic resonance metabolomics in cardiovascular epidemiology and genetics. Soininen P, Kangas AJ, Wurtz P et al.  Circ Cardiovasc Genet. 2015;8:192-206 – Ref 31 in the manuscript). The metabolomic analyses were carried out in two batches, however there are no discernible batch effects with this platform due to the quantitative output of NMR. This is verified by 2 internal QC samples within runs as discussed in Soininen et al. We have added a brief note of this in the text (lines 173 –174).

Point 8: Figures: Font size in figures 1 and 2 should be increased to improve readability. Please specify in the captions if the open blue circles represent uncorrected p-values?
Response 8: We have increased the font size in figures (1,2,3) as suggested to improve readability. The closed blue circles represent significant FDR-corrected p values of <0.05 while the open circles represent non-significant associations. 
Point 9: Line 305-307: it should be noted that the prediction model for GDM risk on which this analysis is based has poor sensitivity and thus, many women from the UPBEAT trial who were indeed at risk for GDM were not included. Also, the specificity of the prediction tool is decent but could be higher, suggesting that some women could have been inappropriately classified as high risk for GDM.

Response 9: We hope that our modifications have now clarified that this paper is not examining the prediction tool. It is in fact examining whether metabolic differences can be found within a high-risk group identified using a prediction tool, when stratified by those who developed GDM and those who did not.

Yours sincerely, 

Ola F Quotah

Reviewer 2 Report

Quotah et al, present a manuscript investigating the “metabolic” profiling of pregnant women with obesity to predict those that will develop GDM. The authors use a predictive algorithm to predict women at risk of developing GDM and carry out the study based upon these findings. They show the metabolic profiles at two gestational ages for those who developed GDM vs those that did not. I think the study is very interesting and would be interesting to see if other diseases can use similar predictive models as it really opens up the possibility for early disease specific markers to be uncovered for either better mechanistic insights, improved diagnostics or therapeutic interventions. However, I suggest a full proof reading and probably rewriting of the results section, this is very difficult to follow and relative concentrations, fold changes, p-values are not discussed at all which I feel harms the manuscript. I would also like to see a more in-depth description of the methods applied. However, with this addressed I think the work will be of interest to the community and readers of metabolites.

Introduction

Page 1, lines 41/42: “increase in postprandial glucose….” I assume this mean circulating? As in blood? Can this just be clarified for readability.

2.1 Participant characteristics

Page 2, lines 94/95: “of whom 119 women (51.5%) developed GDM” it is not clear if the samples were taken based upon the predictions, can this be clarified. If they are taken and then patients later diagnosed with GDM it would be useful to include an average time until diagnosis. Would it be possible to pull out findings based upon time until diagnosis? For examples if there was a range of 2 weeks to 6 months to diagnosis can you see a difference?

Metabolic profiles

Page 4, lines 117/118: I am no expert in gestation and birthing only on the metabolomics and analytical chemistry side so maybe this is an odd question. What does it mean with the 15^+0 and 18^+6? This potentially needs better explanation.

Page 4, lines 130-135: this is a difficult passage to follow would it be possible to show as a figure or table instead maybe? This all seems to be correlations but no mention of concentration fold changes or statistical analysis which would be much more informative and intuitive to the reader. Could this be carried out and added for clarity. It seems from the figure some statistics have been performed but these are not discussed at all in the manuscript.

Page 8, lines 207-223: Most of this is refereeing to SHBG and adiponectin which are protein hormones which are not metabolites so it’s a little confusing as to why they are in a metabolomics paper.

Materials and methods

What were the methods for apolipoprotein classification? Was this done my NMR too? It would also be nice to see some spectra in the SI. Is it similar to that published by Nicholson in 2018 https://doi.org/10.1021/acs.analchem.8b02412 ?

Figures 1-3: this style of figure is unusual for the field so may benefit from additional explanation.

Author Response

Reviewer 2 
We wish to thank the reviewer for their time in reviewing the manuscript.

Comments: 
Point 1: Page 1, lines 41/42: “increase in postprandial glucose….” I assume this mean circulating? As in blood? Can this just be clarified for readability.

Response 1: We have added this to the text for clarification (line 52)

Point 2: Page 2, lines 94/95: “of whom 119 women (51.5%) developed GDM” it is not clear if the samples were taken based upon the predictions, can this be clarified. If they are taken and then patients later diagnosed with GDM it would be useful to include an average time until diagnosis. Would it be possible to pull out findings based upon time until diagnosis? For examples if there was a range of 2 weeks to 6 months to diagnosis can you see a difference?

Response 2: This was a secondary analysis of a prospectively recruited pregnant cohort. Research samples were taken from women at pre-specified timepoints in the study – time point 1 (~16 to 18 weeks’ gestation), timepoint 2, (~24 – 28 weeks’ gestation), as well as a further timepoint not relevant to this analysis.  Oral glucose tolerance tests were undertaken as part of the study in all women at timepoint 2.  The prediction tool was constructed using collected sample data after completion of the UPBEAT study.

Point 3: Page 4, lines 117/118: I am no expert in gestation and birthing only on the metabolomics and analytical chemistry side so maybe this is an odd question. What does it mean with the 15^+0 and 18^+6? This potentially needs better explanation.
Response 3: This is the conventional way of demonstrating weeks plus days during pregnancy. 15 weeks plus 0 days is annotated 15+0, and similarly 18+6 means 18 weeks plus 6 days. We have added ‘gestational weeks’ to the sentence and hope that this adds sufficient clarity. 
Point 4: Page 4, lines 130-135: this is a difficult passage to follow would it be possible to show as a figure or table instead maybe? This all seems to be correlations but no mention of concentration fold changes or statistical analysis which would be much more informative and intuitive to the reader. Could this be carried out and added for clarity. It seems from the figure some statistics have been performed but these are not discussed at all in the manuscript.
Response 4: This information is present in the figures and tables presented – apologies for the lack of clarity. We have now added reference to the appropriate tables/figures through the text for ease of reading. 
Point 5: Page 8, lines 207-223: Most of this is refereeing to SHBG and adiponectin which are protein hormones which are not metabolites so it’s a little confusing as to why they are in a metabolomics paper.
Response 5:  We have included conventional biochemical assays in addition to the NMR metabolome to support this analysis. 
Point 6: What were the methods for apolipoprotein classification? Was this done my NMR too? It would also be nice to see some spectra in the SI. Is it similar to that published by Nicholson in 2018 https://doi.org/10.1021/acs.analchem.8b02412 ?
Response 6: Apolipoproteins were classified using the NMR methodology as described in Quantitative serum nuclear magnetic resonance metabolomics in cardiovascular epidemiology and genetics. Soininen P, Kangas AJ, Wurtz P et al.  Circ Cardiovasc Genet. 2015;8:192-206 and Ref 31 in the manuscript. 
Point 7: Figures 1-3: this style of figure is unusual for the field so may benefit from additional explanation.
Response 7: We appreciate that this style may be unusual in the context of general metabolomics. We feel that it demonstrates the results succinctly. We followed this method previously and it has been used in other publications focusing on GDM metabolomics: – Huhtala, M. S. et al. BMJ Open Diabetes Research and Care. 2021;9(1) – Mokkala, K. et al. Journal of Nutrition.2020;150(1):31-37 – White, S. L. et al. Diabetologia. 2017;60(10):1903-1912 – White, S. L. et al. PLOS ONE. 2020;15(4).

Yours sincerely, 

Ola F Quotah

Round 2

Reviewer 1 Report

The authors have mostly addressed the original comments. However, their response to comment #9 did not include any modifications to the manuscript. I understand that the point of this paper is not to examine the original prediction tool, but since the data being presented relies on selection of cases using that prediction tool, it should be acknowledged as a limitation that the original tool has weaknesses and may have misclassified some participants from the original cohort.

Author Response

Response : The reviewer is of course absolutely correct that the tool misclassifies some individuals. However, this is the crux of the analysis; we sought to explore which metabolic differences exist between individuals identified as high risk via the tool, but stratified by development of GDM, in an attempt to better understand the pathophysiological pathways to hyperglycaemia and potentially improve the predictive tool. We are not sure therefore that the performance of the tool is a limitation for this analysis but agree that clarification is necessary. We have therefore removed the last sentence of the 4th paragraph of the introduction, and the first two sentences of the 5th paragraph, as well as rewriting the last two sentences of the 5th paragraph. We very much hope that this is now clear.

Reviewer 2 Report

I think the authors have done a good job of responding to my queries and have presented an improved manuscript.

The only thing I havea  drawback on is that for the NMR work they refer to other papers for methodology. This is somthing I find frequently doesnt account for study to study variations and thus is not always entiely accurate so I would prefer to see the methods for NMR desribed for this study in the supplemental.

Other than this, I belive the article will be of great interest to the readers and Metabolites and the broader metabolomics and diabetes communities.

Author Response

Response: We have now added a brief description of the NMR methodology to the supplemental data.
